# An Updated *PAH* Mutational Spectrum of Phenylketonuria in Mexican Patients Attending a Single Center: Biochemical, Clinical-Genotyping Correlations

**DOI:** 10.3390/genes12111676

**Published:** 2021-10-23

**Authors:** Marcela Vela-Amieva, Miguel Angel Alcántara-Ortigoza, Isabel Ibarra-González, Ariadna González-del Angel, Liliana Fernández-Hernández, Sara Guillén-López, Lizbeth López-Mejía, Rosa Itzel Carrillo-Nieto, Leticia Belmont-Martínez, Cynthia Fernández-Lainez

**Affiliations:** 1Laboratorio de Errores Innatos del Metabolismo y Tamiz, Subdirección de Medicina Experimental, Instituto Nacional de Pediatría, Secretaría de Salud, Ciudad de Mexico 04530, Mexico; dravelaamieva@yahoo.com (M.V.-A.); sara_guillen@hotmail.com (S.G.-L.); lizbeth712@hotmail.com (L.L.-M.); rosecarrillo.ic@gmail.com (R.I.C.-N.); leticia.belmont@gmail.com (L.B.-M.); 2Laboratorio de Biología Molecular, Subdirección de Investigación Médica, Instituto Nacional de Pediatría, Secretaría de Salud, Ciudad de Mexico 04530, Mexico; malcantaraortigoza@gmail.com (M.A.A.-O.); ariadnagonzalezdelangel@gmail.com (A.G.-d.A.); dralilianafernandez@gmail.com (L.F.-H.); 3Unidad de Genética de la Nutrición, Instituto de Investigaciones Biomédicas, UNAM/Instituto Nacional de Pediatría, Ciudad de Mexico 04530, Mexico; icig@unam.mx

**Keywords:** phenylketonuria, rare diseases, tetrahydrobiopterin, phenylalanine, newborn screening, Latin America, *PAH* molecular spectrum

## Abstract

Establishing the genotypes of patients with hyperphenylalaninemia (HPA)/phenylketonuria (PKU, MIM#261600) has been considered a cornerstone for rational medical management. However, knowledge of the phenylalanine hydroxylase gene (*PAH)* mutational spectrum in Latin American populations is still limited. Herein, we aim to update the mutational *PAH* spectrum in the largest cohort of HPA/PKU Mexican patients (*N* = 124) reported to date. The biallelic *PAH* genotype was investigated by Sanger automated sequencing, and genotypes were correlated with documented biochemical phenotypes and theoretical tetrahydrobiopterin (BH_4_) responsiveness. Patients were biochemically classified as having classic PKU (50%, 62/124), mild PKU (20.2%, 25/124) and mild HPA (29.8%, 37/124). Furthermore, 78.2% of the included patients (97/124) were identified by newborn screening. A total of 60 different pathogenic variants were identified, including three novel ones (c. 23del, c. 625_626insC and c. 1315 + 5_1315 + 6insGTGTAACAG), the main categories being missense changes (58%, 35/60) and those affecting the catalytic domain (56.6%, 34/60), and c. 60 + 5G > T was the most frequent variant (14.5%, 36/248) mainly restricted (69.2%) to patients from the central and western parts of Mexico. These 60 types of variants constituted 100 different biallelic PAH genotypes, with the predominance of compound-heterozygous ones (96/124, 77%). The expected BH_4_ responsiveness based on the *PAH* genotype was estimated in 52% of patients (65/124), mainly due to the p. (Val388Met) (rs62516101) allele. Instead, our study identified 27 null variants with an allelic phenotype value of zero, with a predominance of c. 60 + 5G > T, which predicts the absence of BH_4_ responsiveness. An identical genotype reported in BIOPKUdb was found in 92/124 (74%) of our patients, leading to a genotype–phenotype concordance in 80/92 (86.9%) of them. The high number of variants found confirms the heterogeneous and complex mutational landscape of HPA/PKU in Mexico.

## 1. Introduction

Phenylketonuria (PKU; MIM #261600) is an autosomal recessive disorder caused by the deficiency of the enzyme phenylalanine hydroxylase (PAH; E.C.1.14.16.1), which is encoded by the phenylalanine hydroxylase gene (*PAH,* MIM * 612349; 12q23.2) [1]. To date, more than 1200 pathogenic variants have been described in the international database of patients and genotypes causing hyperphenylalaninemia (HPA)/PKU (BIOPKUdb, www.biopku.com, accessed on 26 August 2021), including those genotypes responsive to tetrahydrobiopterin (BH_4_). PKU prevalence varies worldwide, with an average of 1:10,000 newborns (NB); this is more frequent in Italy (1:2700 NB) and Ireland (1:4500 NB), but it is extremely rare in Thailand (1:212,535 NB) and Japan (1:120,000 NB) [2,3]. In Mexico, the HPA/PKU birth prevalence has been estimated to be 1:27,546 [4]. The main biochemical characteristic of PKU is the elevation of phenylalanine (Phe) concentrations in biological fluids, with a concomitant decrease in tyrosine (Tyr), causing brain damage clinically manifested as intellectual disability, seizures, movement disorders, psychiatric and skin problems, and others [2,5].

Recently, an extensive worldwide description of the *PAH* molecular landscape has been provided, showing the mutational spectra in PKU/HPA patients from different populations [2,3]. Although data from Mexico are included, the reported sample is small (48 patients) and does not contain individuals from all the states that make up the country [6].

According to the recent European and US guidelines for PKU management, the characterization of the responsible *PAH* genotype must be performed in all patients diagnosed with PKU/HPA, which also should be correlated mainly with the expected biochemical phenotype, dietary Phe tolerance and the BH_4_ responsiveness [7,8]. The aim of this study was to present an update to the mutational spectrum of *PAH* in the largest cohort to date of clinically described Mexican PKU patients followed at a single center, showing the genotype/phenotype correlation, with emphasis on the severe c. 60 + 5G > T (rs62514895) founder variant, which is considered to be the most common pathogenic allele in our population [6,9]. Furthermore, herein, we reported three novel variants; in addition, in silico modelling analysis was performed to evaluate the recently described p. (His264Arg) variant (BIOPKUdb) in order to predict its possible pathogenic effect.

## 2. Materials and Methods

### 2.1. Ethics Statement

This study was approved by institutional review boards (2020/014), and written informed consent was obtained from all the participants or their parents. After genotype establishment, all the families received genetic counseling.

### 2.2. Subjects

A total of 142 non-related Mexican patients identified with HPA attending the National Institute of Pediatrics were invited to participate. A scheme workflow is shown in Figure 1, and only the 124 patients bearing biallelic *PAH* genotypes were included. The geographical origin of participants included patients from 30 out of the 32 states in the country. Clinical and demographic data, including the modality of HPA/PKU diagnosis, either by early detection through newborn screening (NBS) or late clinical diagnosis (CD) were registered. As the c. 60 + 5G > T is the most prevalent pathogenic variant in Mexico [6], its clinically and biochemically related phenotype was described, including brain nuclear magnetic resonance imaging (NMRI), in two homozygous and CD patients. The observed biochemical phenotype of patients (67 males and 57 females) was classified following the three categories established by the highest untreated Phe blood concentration as follows: classical PKU (cPKU; blood Phe >1200 μmol/L), mild PKU (mPKU; blood Phe 600–1200 μmol/L) and mild hyperphenylalaninemia (MHP; blood Phe 120–600 μmol/L) [3].

### 2.3. Amino Acid Quantification

The quantification of Phe and Tyr blood concentrations was performed by high-performance liquid chromatography (HPLC) according to a previously described methodology by Qureshi et al. [10] or by tandem mass spectrometry (MS/MS). The determination of blood amino acid profiles by tandem mass spectrometry was performed using a commercial kit from Perkin Elmer (Neobase kit, Wallac Oy, Turku, Finland), according to the manufacturer’s instructions.

### 2.4. Genotype Analysis

Genomic DNA samples were obtained from dried blood spots by the saline precipitation method (Gentra Puregene Blood Kit, Gentra Systems, Minneapolis, MN, USA). Polymerase chain reaction (PCR) amplification and direct automated Sanger sequencing were applied to the thirteen *PAH* exons (NG_008690.2 RefSeqGene, NM_000277.3) in addition to their exon–intron borders. PCR oligonucleotides and sequencing conditions are presented in Appendix A and Appendix B. Identified variants were classified according to BIOPKUdb and/or the literature. Novel variants were classified according to the scoring proposed by the American College of Medical Genetics and Genomics and the Association for Molecular Pathology (ACMG/AMP) [11]. Due to limited available information regarding their functional or clinical relevance, these novel and missense changes were subjected to in silico analysis using the PolyPhen (http://genetics.bwh.harvard.edu/pph2, accessed on 20 December 2020), PROVEAN (http://provean.jcvi.org/index.php, accessed on 20 December 2020), and Pmut (http://mmb.irbbarcelona.org/PMut, accessed on 20 December 2020) programs.

Novel intronic variants were assessed by the splicing module integrated in Alamut^®^ Visual version 2.15 software (SOPHiA GENETICS, Lausanne, Switzerland), which include SpliceSiteFinder-like, MaxEntScan, GeneSplicer and NNSPLICE in silico evaluations for donor and acceptor splice sites, as well as ESEFinder, RESCUE-ESE and EX-SKIP, to predict potentially deleterious effects on Exonic Splicing Enhancer (ESE) binding sites.

### 2.5. Protein Modeling and Mutagenesis in Silico

The crystallographic structure of human PAH with and without ligands (Protein Data Bank (PDB) codes: 1KW0 and 2PAH, respectively) was used to localize and analyze the potential pathogenic effect of the recently reported *p.* (His264Arg) variant (BIOPKUdb) by mutagenesis in silico. Pymol software, version 2.3.5, was used for protein analyses and figure construction [12]. 1KW0 crystal was obtained in the presence of BH_4_ cofactor and the substrate analogue 3-(2-thienyl)-L-alanine (THA) [13].

### 2.6. Genotype–Phenotype Correlation with GPV

Our observed biochemical phenotype (cPKU, mPKU and MHP) was compared with the Genotype Phenotype Value (GPV) calculated from Allelic Phenotype Value (APV) [14] reported in BIOPKUdb. A genotype–phenotype correlation was considered concordant when the obtained GPV corresponded with the reported cut-off.

### 2.7. Genotype–Phenotype Correlation with Identical Genotypes

In the case in which identical genotypes were available in the BIOPKUdb, their associated phenotypes were compared with those found in the patients of the present work. For new variants not reported in BIOPKUdb, the category “not reported” or “still undetermined” was used.

### 2.8. Theoretical BH_4_ Responsiveness and Recommendation to Test

BIOPKUdb was used to analyze the BH_4_ responsiveness related to the genotypes found in the present work. To that end, identical genotypes were investigated. In cases in which genotypes or their BH_4_ responsiveness had not yet been reported in BIOPKUdb, the analysis was performed based on each allele in homozygous state. Since it is well known that genotypes with a GPV equal to zero are non-BH_4_ responders [15,16], genotypes with these characteristics were excluded from the analysis. If the patient contained any allele that had been classified as a responder or a slow responder in BIOPKUdb, the recommendation was to test for BH_4_ responsiveness. This same recommendation to test was also made for all the genotypes that had not yet been reported in BIOPKUdb.

### 2.9. Statistical Analysis

Descriptive statistics (average, minimum and maximum) were determined for clinical data of patients. Allelic frequencies were calculated as the number of alleles of each variant per 100/total number of studied alleles. Comparisons of frequencies were performed using a Chi-squared test with MedCalc software version 20.010 (MedCalc Software Ltd., Ostend, Belgium) considering a *p* value < 0.05 as significant.

## 3. Results

From the 124 included patients, 97 (78.2%) were early detected by NBS, and 27 (21.8%) belonged to the CD group. Based on the Phe blood concentrations, we observed that 62/124 cases were cPKU (50%), 25/124 were mPKU (20.2%) and 37/124 were MHP (29.8%).

### 3.1. PAH Variants

The identified biallelic *PAH* genotypes included a total of 60 different pathogenic variants, the most frequent being c. 60 + 5G > T among the 124 patients (36/248 alleles, 14.5%). The frequency and type of these 60 different pathogenic variants are shown in Table 1. Six of them had a statistically significant difference of frequency compared to that reported in BIOPKUdb (Table 1). Most pathogenic changes accounted for missense changes (58%), affected the catalytic domain (56.6%) (Figure 2) and involved exon 7 (13, 21.7%, Figure 3).

### 3.2. Genotypes

We found 100 different biallelic genotypes, most of them (96/124, 77%) in compound heterozygous state (Table 2). Regarding homozygous genotypes, the c. [60 + 5G < T]; [60 + 5G > T] was the most frequent one (10/28 patients), followed by the c. [1162G > A]; [1162G > A] genotype (5/28 patients). From the total of 100 genotypes, 11% of them included the c. 60 + 6G > T variant, either in homozygous or compound heterozygous state, in which c. [60 + 5G > T]: [1162G > A] (4 cases) and c. [60 + 5G > A]: [441 + 5G > T] (3 cases) predominate for this last category. 

### 3.3. Genotype/Phenotype Concordance with GPV

Concordance with the GPV was established in 118/124 cases (95.2%). In 6/124 (4.8%) cases, concordance could not be established as the allele was reported as undetermined in the BIOPKUdb. From the remaining 118 cases with information available, 85 were concordant (72%) and 33/118 cases were discordant (28%).

### 3.4. Genotype/Phenotype Concordance with Identical Genotypes Reported in BIOPKUdb

An identical genotype reported in BIOPKUdb was found in 92 out of 124 patients (74%). From those 92 subjects, 80 (86.9%) were concordant with the reported phenotype, and in 12 cases, the phenotype was discordant. The disease severity was under- or overestimated in eight and four cases, respectively (Table 2).

### 3.5. Clinical Description of Patients Bearing PAH Genotypes Containing c. 60 + 5G > T Variant

This pathogenic variant alters the consensus sequence of the 5’ donor splice site of intron 1 of the *PAH* gene [17], possibly leading to the abolition of its proper recognition by splicing machinery. A total of 26 patients carried the c. 60 + 6G > T variant, either in homozygous (10/26) or in compound heterozygous (16/26) state (Table 3), and 18/26 (69.2%) of them came from the Mexican Middle West region called El Bajío (Jalisco, Guanajuato, Michoacán and Aguascalientes States). Sixty five percent of these patients (17/26) were detected early by NBS, while the remaining patients belonged to the CD group. Brain NMRI imaging of two c. 60 + 5G > T-homozygous patients from the CD group demonstrated damage in the basal ganglia, white matter, and cerebral atrophy (Figure 4). The first patient was a one-year-old male who was detected at 9 months old; his Phe diagnostic value was 1654 μM, he was clinically diagnosed with autism and severe neurodevelopmental delay, and he had poor outcome after nutritional therapy. The other patient was a female who had a diagnostic Phe value of 1830 μM, at 10 years old, with severe neurodevelopmental delay and behavioral issues, such as auto-aggression and incontrollable tantrums. She was misdiagnosed with autism. After PKU diagnosis, nutritional therapy was started with good compliance and biochemical Phe controls; however, only a few clinical benefits were observed after treatment, such as a decrease in auto-aggression and tantrums.

### 3.6. Novel PAH Variants and Description of Resulting Phenotypes

The frameshift c. 23del or p. (Asn8Thrfs * 30) variant was found in a patient categorized as cPKU and bearing the p. (Val388Met) in the second allele (Patient 46, Table 2). Clinically, this patient showed a maximum blood Phe concentration of 2798 μmol/L, a Tyr blood concentration of 133.29 μmol/L, a Phe/Tyr ratio of 20.9 and a dietary Phe tolerance of 215 mg/day. This patient began nutritional treatment at 1.5 months old, with consistent adherence to treatment during 4 years of follow up, with most of her Phe values within the therapeutical range (<360 μmol/L) and neurodevelopmental skills according to her age.

The c. 1315 + 5_1315 + 6insGTGTAACAG variant was identified in compound heterozygosity involving the p. (His264Arg) variant (Patient 88, Table 2) in a cPKU patient showing Phe blood levels of 1753 μmol/L, Tyr of 75 μmol/L, a Phe/Tyr ratio of 23.37 and dietary Phe tolerance of 255 mg/day. The patient initiated nutritional treatment at the neonatal stage, and during a follow-up period of 5 years, he displayed normal neurodevelopmental skills according to his age. This intronic insertion is located at the 5′ end of intron 12 of *PAH*, and although it does not directly disturb the canonical donor splice site, the in silico evaluation agrees with the diminished recognition of the intron 12 donor splice site (MaxEntScan: −42.9%, NNSPLICE: −8.8%, SpliceSiteFinder-like: −6.4%, GeneSplicer: −100%; ESEFinder: abolition of a splicing intronic enhancer element recognized by SRp55).

The homozygous c. [625_626insC]; [625_626insC] genotype for a novel frameshift insertion was identified in a cPKU patient (Patient 72, Table 2), showing a Phe blood concentration of 1217 μmol/L, Tyr of 58 μmol/L, a Phe/Tyr ratio of 20.9 and a dietary Phe tolerance of 319 mg/day. As this patient was adopted at an early age, we did not have his family history (i.e., parental consanguinity). Early instauration of dietary management at the age of 25 days allowed him to maintain most of his blood Phe values within the therapeutic range and normal neurodevelopmental skills according to his age, in a 6.5-year follow-up period. 

### 3.7. Protein in Silico Modeling of the p. (His264Arg) Variant

The c. 791A > G or p. (His264Arg) has been recently reported in the BIOPKUdb, although its allele frequency is still not known, results of functional assays are not available, and its resulting biochemical phenotype is still unknown. In fact, in the ClinVar database, it is considered as a variant of unknown significance (RCV001224584.2). Herein, it was identified in two unrelated patients (ID 64 and 88, Table 2). The PolyPhen, PROVEAN, and Pmut programs unanimously predicted the p. (His264Arg) variant as deleterious with high confidence scores. Due to the rarity of this variant, we evaluated in silico the potential disturbance at the tridimensional structural arrangement of the resulting protein. We found that the substitution of histidine by the positively charged amino acid arginine predicts a serious disturbing effect within the vicinity of the catalytic site in all the possible rotamers of arginine (Figure 5).

### 3.8. Theoretical BH_4_ Responsiveness

We found that there were 52% (65/124) of patients whose genotypes included at least one potential responder allele, or one that has still not been reported in BIOPKUdb (Table 4). The MHP was the predominant biochemical phenotype of these patients (52.3%, 34/65), but cPKU accounted for 27.7% (18/65), and the remainder accounted for mPKU (20%, 13/65). The most frequent responder allele was c. 1162G > A or p. (Val388Met) (14 alleles), followed by c. 782G > A or p. (Arg261Gln) (seven alleles) and c. 722G > A or p. (Arg241His) and c. 1169A > G or p. (Gln390Gly) (six alleles each).

## 4. Discussion

The epidemiology of PKU has been the objective of a growing number of publications [2,3,18]; however, the main findings of those investigations are based on European and Asiatic populations. The Latin American population is poorly represented, mainly due to the scarce reliable published data from this region. In a recent PKU work, genetic and epidemiological data from 64 countries were presented [2]; however, only three Latin American countries were considered (Argentina, Brazil and Mexico [6,19,20]), so there would be a possible underrepresentation of the mutational *PAH* spectrum in Latin American HPA/PKU patients. The present study aimed to update the *PAH* mutational spectrum of Mexican HPA/PKU patients attending a single reference center. Moreover, to the best of our knowledge, this is the largest published cohort of PKU patients from Mexico, including individuals of almost all the geographic regions of the country (patients came from 30/32 of the states that constitute the country).

In this new study, patients’ phenotypes were classified according to the criteria established by Hillert et al. and van Spronsen et al. [2,3], which will facilitate international comparisons both with BIOPKUdb and other publications. By comparing the frequencies of the different *PAH*-related biochemical phenotypes identified herein with those previously reported [6], no statistical differences were found.

The number of patients detected by NBS in our center increased from 54.1% in 2015 [6] to 78.2%. This significant increment (*p* = 0.001) could be related to the optimization of the NBS system in Mexico, although it still does not reach the optimal goal of having a total NBS coverage for this disease [21]. This finding agrees with the fact that improvements in the NBS programs increase the detection of MHP forms [22,23]. Even though it is widely accepted that MHP patients are only treated when Phe blood levels are >360 μmol/L, the biochemical follow up of these patients should be periodical, in order to assure neurologically safe Phe values [24]. Moreover, in female patients at a reproductive stage, a close follow up is especially relevant considering the potential risk of their offspring suffering from maternal PKU syndrome [2].

As reported has been worldwide [2,3], the molecular PKU spectrum found in the present study is heterogeneous, and variants are distributed along the entire gene with clustering in exons that codify the catalytic PAH region (35%), especially exons 7 and 6 (Figure 3). Notably, we did not find any patient with the “Celtic” c. 1222C > T or p. (Arg408Trp) pathogenic variant, which is considered the most frequent one worldwide, especially in central and eastern Europe (44.4–53.7%); however, its allele frequency in the Spanish population decreased significatively (4%), which could correlate with its absence in the Mexican population, which is characterized by bearing a Spanish-derived ancestry [3]. On the other hand, the absence of variants in exon 13 (Figure 3) was similar to that observed in Spain and other Latin American populations [19,25,26,27,28]. This fact is in agreement with the low number of pathogenic variants registered in exon 13 in BIOPKUdb.

The Sanger methodology allowed us to obtain a detection rate of 87.3% (124/142) of the biallelic *PAH* genotypes in the patients initially classified as HPA/PKU; however, this methodology identified that there were 2.81% (4/142) of patients carrying only one pathogenic allele (Figure 1). To date, there are no available data to estimate the proportion of large rearrangements unnoticed for Sanger sequencing methodology, which are expected in less than 1% of *PAH*-related HPA patients worldwide [3]. At least in Russian patients, gross exonic deletions (mainly involving exon 3 and 5) identified by Multiplex Ligation-dependent Probe Amplification (MLPA) analysis account for 0.39% of pathogenic *PAH* alleles [29]. Thus, in these four patients with monoallelic genotypes—who presented persistent HPA, three of which required nutritional treatment—further studies, such as MLPA analysis, could be an option in order to discard gross *PAH* rearrangements.

The remaining 14 cases (9.8%) with normal *PAH* genotypes lacked the recommended former dihydropteridine reductase (DHPR) and pterin evaluations [2] to discard a defect in the metabolism of BH_4_.

Our study corroborates the high frequency of the c. 60 + 5G > T variant (14.5%, Table 1) and its geographical predominance in the western and central regions of Mexico (Table 2), probably due to the founder effect [6]; this has been corroborated by other authors [9]. This intronic null variant, formerly known as “IVS1 + 5g > t”, was first described by Guldberg et al. in 1993 [30]. It is a rare variant reported in only 0.7% and 1.36% of Danish [30] and Spanish [31] populations, respectively. In BIOPKUdb, it is reported in 0.32% of the subjects. Due to its rarity, it has been scarcely studied. Interestingly, this variant has not been found in PKU patients from Argentina, Chile or Cuba [19,27,32], but it has been reported in Costa Rica [33], and Brazil [20,25,34,35]. Moreover, the c. 60 + 5G > T variant has not been reported in Russia [36], Japan [23], or China [37]. Therefore, to the best of our knowledge, the studied Mexican population has the highest frequency of this variant.

In the present study, all the patients harboring a homozygous c. 60 + 5G > T genotype had cPKU, which can be explained by the severity of this null variant (APV value = 0), which can theoretically lead to no enzymatic activity, in addition to an absence of BH_4_ response [15]. As expected, among patients harboring c. 60 + 5G > T in compound heterozygosity involving a less severe allele with APV > 5 [14], the disease was less severe, corresponding to an MHP phenotype (patients 23 and 24, Table 3), confirming that functionally mild variants with a substantial residual PAH enzymatic activity dominate over null alleles [3]. The clinical picture of patients with homozygous genotypes c. [60 + 5G > T]; [60 + 5G > T] coincides with that described for cPKU patients, consisting of early, severe and progressive neurological manifestations, and symptomatology worsening if the start of dietary treatment is delayed. Therefore, these findings emphasize the importance of starting treatment in the first days of life, preferably in the first week [8], to prevent brain damage that leads to neurodevelopmental delay and permanent intellectual disability [38]. At least in two c. 60 + 5G > T-homozygous and lately diagnosed patients (CD group), we documented central nervous system sequelae (Figure 4**)**, consisting of basal ganglia and white matter brain damage. Remarkably, although the male patient was diagnosed at a younger age, he showed the involvement of basal ganglia, but it is known that imaging studies of PKU patients do not always correlate with the severity of the observed phenotype [39].

The second most frequent variant of this work was c. 1162G > A p. (Val388Met) (11.2%). This variant is common in the Spanish population (6.8%) [26], as well as in other Latin American countries such as Chile (17.2%) [27], Argentina (3.9%) [19] and Brazil (12.7–21.2%) [25,28]. Interestingly, our five homozygous p. (Val388Met) patients had different biochemical phenotypes (mPKU: *N* = 4 and cPKU: *N* = 1, Table 2). The predominance of mPKU phenotypes observed herein associated with homozygous p. (Val388Met) contrasts with that reported in BIOPKUdb, where half of the patients show cPKU. Furthermore, this variant was the most frequently found in the discordant heterozygous genotypes (4/12) and in the potential BH_4_-responder genotypes (13/60, Table 4). Its predominance among patients with an inconsistent phenotype could be related to diverse proposed phenomena, such as its destabilizing PAH nature; differences in protein folding; or other, less known factors, such as the influence of modifier genes or epigenetic modifications [26].

The c. 441 + 5G > T or p. (?) intronic and severe variant (APV = 0), was found in 5.64% of the studied alleles, which contrasts with its low frequency worldwide (0.98%). However, higher allelic frequencies have been reported for other Latin American countries, such as Argentina (2.4%) [19] and Chile (3.0%) [27]. Severe cPKU is expected for those homozygous individuals bearing this null allele. Remarkably, in the present study, 50% of the patients with this variant, including two homozygous patients, are originally from the north of Mexico (Baja California, Chihuahua, Coahuila and Tamaulipas). This likely suggests a possible founder effect similar to that described for c. 60 + 5G > T [6]; therefore, further research is needed to support this hypothesis.

The c. 1066-11G > A or p. (Gln355_Tyr356insGlyLeuGln) variant has an APV = 0 leading to the cPKU phenotype. It was identified in 5.6% of the alleles (Table 1), which is similar to that reported in BIOPKUdb (6.8%), Argentina (9.3%), Brazil (9.4%) [3] and Chile (12.7%) [27]. This variant was previously proposed as being responsible for the high PKU cases from Jalisco State [40]; however, our work does not support this previous asseveration, since we did not observe any geographical predominance for patients carrying this variant.

Regarding the novel c. 1315 + 5_1315 + 6insGTGTAACAG variant, it is currently classified as likely pathogenic, as it meets the PM2, PM3, PP3 and PP4 ACMG/AMP criteria [11]. The paternal origin of this variant was confirmed in cPKU Patient #88, who has an extremely low dietary Phe tolerance (255 mg/day) associated with his compound heterozygous genotype involving c. 791A > G or p. (His264Arg) of maternal origin (Table 2). This last missense variant was recently posted in BIOPKUdb, and we predicted it to be a pathogenic change (Figure 5), supported by PM1, PM2, PM5, PP3 and PP4 ACMG/AMP criteria [11].

The novel c. 625_626insC variant was found in homozygous state in patient #77 affected by cPKU (Table 2), in addition to low dietary Phe tolerance (319 mg/day), but no familial antecedents were available as the patient was adopted. This frameshift variant was classified as pathogenic according to the ACMG/AMP (American College of Medical Genetics / Association for Molecular Pathology) criteria (very strong pathogenic variant (PVS1), pathogenic moderate (PM1, PM2) and PP4 ACMG/AMP) [11].

The third novel c. 23del or p. (Asn8Thrfs * 30) frameshift variant was also classified as pathogenic, as it meets the PVS1, PM1, PM2 and PP4 ACMG/AMP criteria [11]. This variant was in heterozygous state with c. 1162G > A or p. (Val388Met) for patient #46 (Table 2), who was classified as a cPKU with low Phe dietary tolerance (215 mg/day). To our knowledge, this variant has never been documented; however, Wang et al., 2018, reported a similar c. 23_24delinsT or p. (Asn8Ilefs * 30) variant (PAH1043 accession number BIOPKUdb) [41]. Although it is a different variant, the resulting frameshift effect at the regulatory domain is the same. The occurrence of these two indels at the same position could suggest a “hot spot” for small rearrangements in c. 23 and c. 24 positions.

In the present work, the observed general genotype/phenotype concordance was 83.8% when compared with identical genotypes reported in the BIOPKUdb. These data are slightly higher than previously reported by our group (70.8%) [6]. This higher concordance is likely due to the new classification criteria used, although other unknown epigenetic or environmental regulatory factors could be involved. One limitation for the frequency of variants and concordance comparison is that not all patients from different populations are listed in the BIOPKUdb there are few identical genotypes, especially those with rare variants, hence the importance of contributing to this database with the information of the *PAH* mutational spectrum from diverse world regions. Interestingly, in this study, discordance consisted mainly of the underestimation of the disease severity (8/12, 66.7%), as we classified patients with a less severe disease type. This could highlight the difficulties of establishing the correct phenotype based only on blood Phe levels and the importance of establishing the genotype in all HPA/PKU patients [42].

Finally, we found that 52% of the patients were considered potential BH_4_ responders (Table 4). This is similar to the worldwide responsiveness reported by Hillert et al., who found 43% responsiveness from a cohort of 5597 subjects [3]. The most frequent potential BH_4_ responder variants are of the missense type, with residual enzymatic activity that could be enhanced with pharmacological doses of BH_4_ [43]. On the other hand, we emphasize that patients carrying two non-responder alleles in trans should not be submitted to the BH_4_ response test, since those variants (27/60, 45%) do not allow the synthesis of a functional protein; thus, the recommendation for those patients is to maintain a strictly nutritional treatment [15,16].

## 5. Conclusions

In summary, the high number of variants found, including the three new variants, confirms the heterogeneous and complex mutational landscape of PKU in Mexico. Half of the studied Mexican patients had cPKU, with a predominance of the c. 60 + 5G > T variant (14.5% of the alleles), which could indicate the highest allelic frequency reported for this variant worldwide. Furthermore, we highlight the severe central nervous system sequelae in two late-diagnosed symptomatic c. 60 + 5G > T-homozygous patients which points out the importance of an early instauration of dietary treatment. A high genotype–phenotype concordance was observed (>80%). A possible founder effect of the c. 441 + 5G > T variant in the north of Mexico warrants further research. 

## Figures and Tables

**Figure 1 genes-12-01676-f001:**
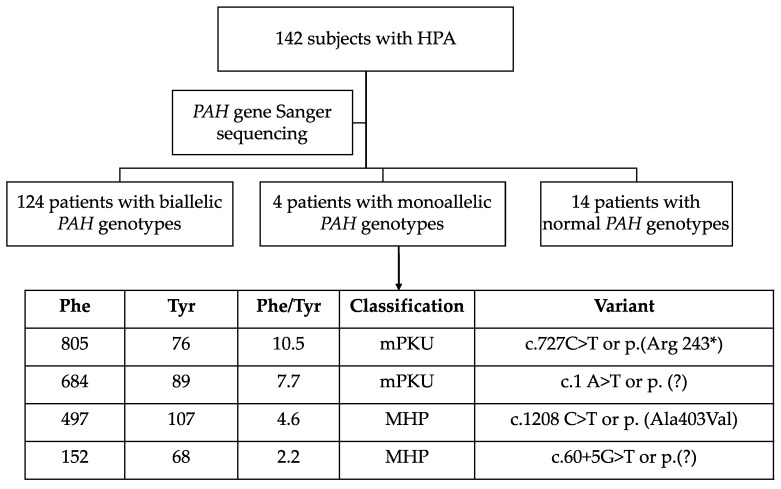
Workflow scheme for inclusion in the present study of 124 patients bearing *PAH* biallelic genotypes. The biochemical characteristics of the four identified patients bearing monoallelic *PAH* genotypes are shown. The 14 patients with normal *PAH* genotypes are currently under study for BH_4_ defects. Abbreviations: HPA: hyperphenylalaninemia; PAH: phenylalanine hydroxylase gene; Phe: phenylalanine; Tyr: tyrosine; mPKU: mild phenylketonuria; MHP: mild hyperphenylalaninemia.

**Figure 2 genes-12-01676-f002:**
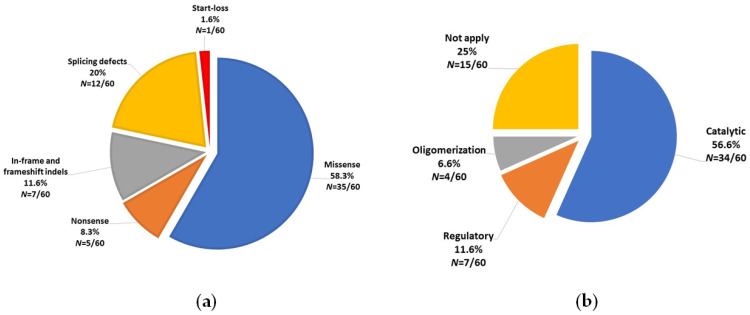
Main categories of the 60 types of different pathogenic variants identified among the 124 unrelated Mexican patients bearing a diagnostic biallelic *PAH* genotype. (**a**) Classification by predicted pathogenic effect of the variant. (**b**) Classification by affected domain. Frameshift, start-loss, splicing defects and non-sense were considered as null alleles. *N* = number of alleles/total of alleles (60).

**Figure 3 genes-12-01676-f003:**
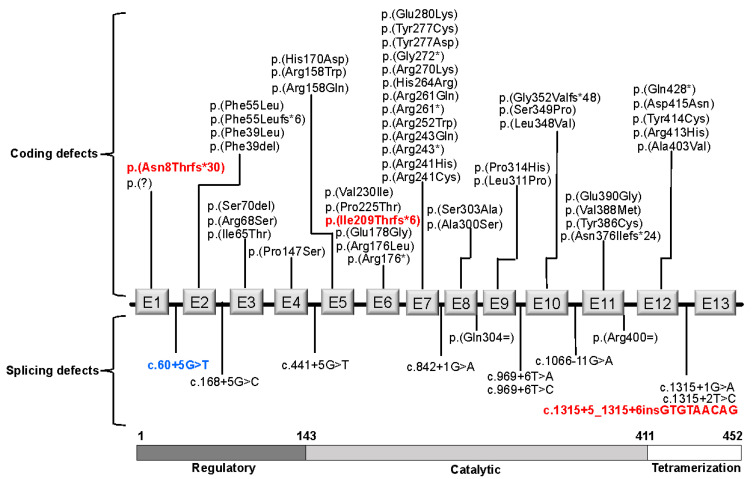
*PAH* gene diagram showing the localization of the 60 types of variants found in the present study. Novel variants are in red bold type. The most common pathogenic c. 60 + 5G > T splicing variant (14.5% of the *PAH* alleles) is highlighted in blue bold type. The apparently synonymous variants, p. (Gln304=) and p. (Arg400=), disrupt an exon splicing enhancer element, leading to a splicing defect (BIOPKUdb). E1-E13 diagrammatic representations of exons 1 to 13 of *PAH*.

**Figure 4 genes-12-01676-f004:**
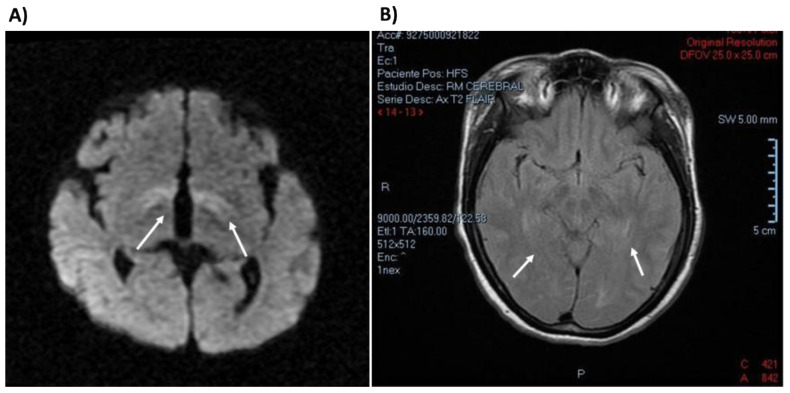
Brain NMRI imaging of two CD patients diagnosed with homozygous genotype c. 60 + 5G > T, and cPKU phenotype. (**A**) Diffusion NMRI of one-year-old male, showing restricted diffusion in periventricular white matter and basal ganglia (arrows). Hypomyelination and brain atrophy were also observed. (**B**) Axial T2 FLAIR imaging of a 10-year-old female showing periventricular white matter hyperintensity (arrows).

**Figure 5 genes-12-01676-f005:**
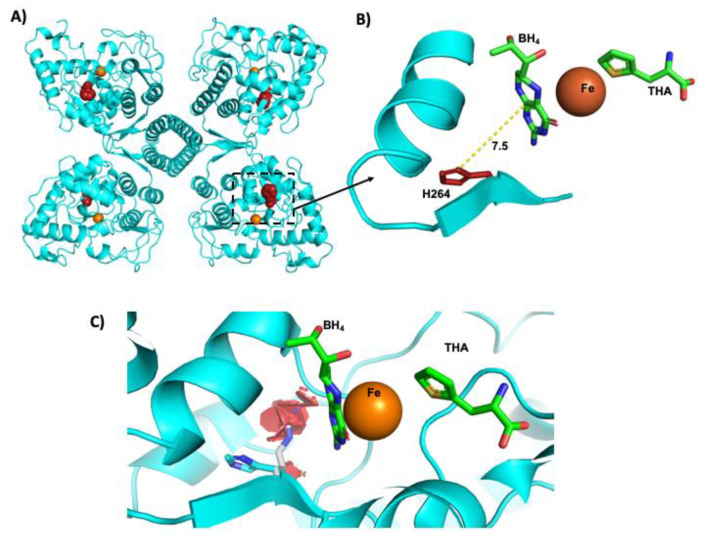
Location of His-264 residue in PAH structure. (**A**) His-264 residue is located within the vicinity of the active site; (**B**) zoom of His-264 location that shows a distance of 7.5 Å from the catalytic PAH site; (**C**) illustration of resulting repulsive clashes (red hexagons) conditioned by the His residue substitution by the positively charged Arg residue. The figure was constructed with Pymol [13]. THA = 3-(2-thienyl)-L-alanine (phenylalanine analogue).

**Table 1 genes-12-01676-t001:** Frequency and type of *PAH* variants found in the studied population (248 alleles) and their comparison and categorization accordingly to information available in BIOPKUdb.

				BIOPKUdb			
Classification According APV ^1^	*PAH* Variant	Protein Change	% (Present Work)	%	APV ^2^	EA (%)	Protein Domain
cPKU (*N* = 42)	c. 60 + 5G > T ▪	p. (?)	14.516 ^♦^	0.320	0.0	NR	NA
	c. 1162G > A	p. (Val388Met)	11.290 ^♦^	1.800	1.9	28	Catalytic
	c. 441 + 5G > T ▪	p. (?)	5.645	0.980	0.0	NR	NA
	c. 1066-11G > A ▪	p. (Gln355_Tyr356insGlyLeuGln)	5.645	6.800	0.0	5	Catalytic
	c. 1045T > C ▪	p. (Ser349Pro)	4.435	1.000	0.0	1	Catalytic
	c. 782G > A	p. (Arg261Gln)	2.823	5.500	1.6	44	Catalytic
	c. 194T > C	p. (Ile65Thr)	2.419	4.000	1.0	33	Regulatory
	c. 809G > A ▪	p. (Arg270Lys)	2.419	0.220	0.0	11	Catalytic
	c. 1A > T ▪	p. (?)	2.016 ^♦^	0.026	0.0	NR	NA
	c. 1315 + 1G > A ▪	p. (?)	2.016	4.400	0.0	NR	NA
	c. 728G > A ▪	p. (Arg243Gln)	1.613	2.700	0.0	14	Catalytic
	c. 838G > A ▪	p. (Glu280Lys)	1.613	1.300	0.0	11	Catalytic
	c. 842 + 1G > A ▪	p. (?)	1.613	0.440	0.0	NR	NA
	c. 1055del ▪	p. (Gly352Valfs * 48)	1.613	0.590	0.0	NR	Catalytic
	c. 208_210delTCT	p. (Ser70del)	1.210	0.026	0.7	NR	Regulatory
	c. 673C > A ▪	p. (Pro225Thr)	1.210	0.170	0.0	NR	Catalytic
	c. 754C > T ▪	p. (Arg252Trp)	1.210	1.400	0.0	15	Catalytic
	c. 781C > T ▪	p. (Arg261 *)	1.210	0.770	0.0	NR	Catalytic
	c. 830A > G ▪	p. (Tyr277Cys)	1.210	0.016	0.0	NR	Catalytic
	c. 969 + 6T > A	p. (?)	1.210 ^♦^	0.005	NR	NR	NA
	c. 1042C > G	p. (Leu348Val)	1.210	0.940	1.5	25	Catalytic
	c. 117C > G	p. (Phe39Leu)	0.806	0.420	0.2	49	Regulatory
	c. 165del ▪	p. (Phe55Leufs * 6)	0.806	0.910	0.0	NR	Regulatory
	c. 439C > T ▪	p. (Pro147Ser)	0.806	0.042	0.0	NR	Catalytic
	** c. 625_626insC **	** p. (Ile209Thrfs * 6) **	0.806	NR	NR	NR	NA
	c. 727C > T ▪	p. (Arg243 *)	0.806	1.200	0.0	NR	Catalytic
	c. 791A > G	p. (His264Arg)	0.806	NR	NR	NR	Catalytic
	c. 1157A > G ▪	p. (Tyr386Cys)	0.806	0.074	0.0	NR	Catalytic
	** c. 23del **	** p. (Asn8Thrfs * 30) **	0.403	NR	NR	NR	NA
	c. 116_118delTCT	p. (Phe39del)	0.403	0.130	0.9	NR	Regulatory

	c. 168 + 5G > C ▪	p. (?)	0.403	0.600	0.0	NR	NA
	c. 472C > T ▪	p. (Arg158Trp)	0.403	0.110	0.0	2	Catalytic
	c. 473G > A ▪	p. (Arg158Gln)	0.403	2.900	0.0	10	Catalytic
	c. 526C > T ▪	p. (Arg176 *)	0.403	0.410	0.0	NR	Catalytic
	c. 814G > T ▪	p. (Gly272 *)	0.403	0.790	0.0	NR	Catalytic
	c. 829T > G	p. (Tyr277Asp)	0.403	0.240	0.4	NR	Catalytic
	c. 932T > C ▪	p. (Leu311Pro)	0.403	0.170	0.0	10	Catalytic
	c. 1127del	p. (Asn376Ilefs * 24)	0.403	0.005	1.0	NR	Catalytic
	c. 1198A > C	p. (Arg400=)	0.403	0.005	NR	NR	NA
	c. 1282C > T ▪	p. (Gln428 *)	0.403 ^♦^	0.000	0	NR	Oligomerization
	c. 1315 + 2T > C ▪	p. (?)	0.403	0.026	0.0	NR	NA
	** c. 1315 + 5_1315 + 6insGTGTAACAG **	** p. (?) **	0.403	NR	NR	NR	NA
mPKU (*N* = 6)	c. 204A > T	p. (Arg68Ser)	1.210 ^♦^	0.600	5.4	25	Regulatory
	c. 721C > T	p. (Arg241Cys)	0.403	1.100	5.5	57	Catalytic
	c. 722G > A	p. (Arg241His)	2.419	0.460	5.2	23	Catalytic
	c. 912G > A	p. (Gln304=)	0.806	0.095	3.3	NR	NA
	c. 969 + 6T > C	p. (?)	0.403	0.005	NR	NR	NA
	c. 1241A > G	p. (Tyr414Cys)	0.806	3.100	5.1	57	Oligomerization
MHP (*N* = 12)	c. 165T > G	p. (Phe55Leu)	0.403	0.130	8.2	NR	Regulatory
	c. 508C > G	p. (His170Asp)	1.613	0.053	10.0	43	Catalytic
	c. 527G > T	p. (Arg176Leu)	0.403	0.150	9.8	42	Catalytic
	c. 533A > G	p. (Glu178Gly)	2.419	0.300	7.6	39	Catalytic
	c. 688G > A	p. (Val230Ile)	0.403	0.270	10.0	63	Catalytic
	c. 898G > T	p. (Ala300Ser)	0.403	1.500	9.7	65	Catalytic
	c. 907T > G	p. (Ser303Ala)	0.403	0.047	9.3	NR	Catalytic
	c. 941C > A	p. (Pro314His)	0.403	0.037	10.0	NR	Catalytic
	c. 1169A > G	p. (Glu390Gly)	2.419	1.300	6.9	62	Catalytic
	c. 1243G > A	p. (Asp415Asn)	3.226	0.440	10.0	72	Oligomerization
	c. 1208C > T	p. (Ala403Val)	2.016	2.400	9.7	66	Catalytic
	c. 1238G > A	p. (Arg413His)	0.403	NR	9.0	NR	Oligomerization

^1^ Based on the registry of 16,270 *PAH* alleles listed in BIOPKUdb. ^2^ APV = allelic phenotype value. Null variants have an APV = 0. Novel variants identified herein are highlighted in red bold type. Abbreviations: cPKU, classic phenylketonuria; EA, enzyme activity as reported in BIOPKUdb; MHP, mild hyperphenylalaninemia; mPKU, mild phenylketonuria; NA, not applicable (splice, non-sense, frameshift, or translation initiation defects); NR, not reported or still undetermined. ^♦^ Statistically significant difference between the frequency of the present work and the one reported in BIOPKUdb. ▪ Represents non-BH_4_ responsive alleles, either in homozygous state or in trans with another non-responder allele.

**Table 2 genes-12-01676-t002:** Genetic spectrum, demographic, clinical, and biochemical characteristics of the studied Mexican HPA/PKU patients (*N* = 124).

Patient ID	*PAH* Genotype	Protein Change	Diagnosis	Sex	Geographical Origin	Max Historical Phe (μmol/L)	Observed Biochemical Phenotype	Genotypic Phenotype Value (GPV)
^♦^ 1	c. [60 + 5G > T]; [60 + 5G > T]	p. [?]; [?]	NBS	M	Jalisco	3591	cPKU	0
^♦^ 2	c. [60 + 5G > T]; [60 + 5G > T]	p. [?]; [?]	CD	F	CDMX	2421	cPKU	0
3	c. [60 + 5G > T]; [60 + 5G > T]	p. [?]; [?]	CD	M	Jalisco	1896	cPKU	0
^♦^ 4	c. [60 + 5G > T]; [60 + 5G > T]	p. [?]; [?]	CD	M	Jalisco	1654	cPKU	0
5	c. [60 + 5G > T]; [60 + 5G > T]	p. [?]; [?]	CD	M	Jalisco	1876	cPKU	0
6	c. [60 + 5G > T]; [60 + 5G > T]	p. [?]; [?]	NBS	F	Jalisco	1573	cPKU	0
7	c. [60 + 5G > T]; [60 + 5G > T]	p. [?]; [?]	NBS	M	Jalisco	1311	cPKU	0
8	c. [60 + 5G > T; 60 + 5G > T]	p. [?]; [?]	NBS	M	Jalisco	1712	cPKU	0
9	c. [60 + 5G > T]; [60 + 5G > T]	p. [?]; [?]	NBS	F	Edo Mex	1581	cPKU	0
10	c. [60 + 5G > T]; [60 + 5G > T]	p. [?]; [?]	NBS	F	Jalisco	1453	cPKU	0
^♦^ 11	c. [60 + 5G > T]; [1162G > A]	p. [?]; [Val388Met]	CD	F	Puebla	1900	cPKU	1.8
^♦^ 12	c. [60 + 5G > T]; [1162G > A]	p. [?]; [Val388Met]	NBS	M	Jalisco	3832	cPKU	1.8
^♦^ 13	c. [60 + 5G > T]; c. [1162G > A]	p. [?]; [Val388Met]	NBS	M	Aguascalientes	1876	cPKU	1.8
^♦^ 14	c. [60 + 5G > T]; [1162G > A]	p. [?]; [Val388Met]	NBS	M	Jalisco	1715	cPKU	1.8
15	c. [60 + 5G > T]; [441 + 5G > T]	p. [?]; [?]	NBS	F	Edo Mex	1682	cPKU	0
^♦^ 16	c. [60 + 5G > T]; [441 + 5G > T]	p. [?]; [?]	NBS	F	Baja California Sur	1815	cPKU	0
^♦^ 17	c. [60 + 5G > T]; [441 + 5G > T]	p. [?]; [?]	CD	M	Jalisco	1346	cPKU	0
^♦^ 18	c. [60 + 5G > T]; [1315 + 1G > A]	p. [?]; [?]	NBS	F	Guanajuato	1385	cPKU	0
19	c. [60 + 5G > T]; [1315 + 1G > A]	p. [?]; [?]	NBS	F	Guanajuato	1206	cPKU	0
20	c. [60 + 5G > T]; [673C > A]	p. [?]; [Pro225Thr]	NBS	F	Yucatan	2326	cPKU	0
^♦^ 21	c. [60 + 5G > T]; [727C > T]	p. [?]; [Arg243 *]	CD	F	Jalisco	832	cPKU	0
^♦^ 22	c. [60 + 5G > T]; [1066-11G > A]	p. [?]; [Gln355_Tyr356insGlyLeuGln]	NBS	F	Jalisco	1498	cPKU	0
23	c. [60 + 5G > T]; [1169A > G]	p. [?]; [Glu390Gly]	NBS	F	Baja California	1616	MHP	6.9
^♦^ 24	c. [508C > G]; [60 + 5G > T]	p. [His170Asp]; [?]	NBS	M	Jalisco	1423	MHP	10
^♦^ 25	c. [754C > T]; [60 + 5G > T]	p. [Arg252Trp]; [?]	CD	M	CDMX	2075	cPKU	0
^♦^ 26	c. [838G > A]; [60 + 5G > T]	p. [Glu280Lys]; [?]	CD	F	Jalisco	1949	cPKU	0
27	c. [1162G > A]; [1162G > A]	p. [Val388Met]; [Val388Met]	NBS	M	Puebla	1900	mPKU	1.8
28	c. [1162G > A]; [1162G > A]	p. [Val388Met]; [Val388Met]	NBS	F	Puebla	1516	mPKU	1.8
29	c. [1162G > A]; [1162G > A]	p. [Val388Met]; [Val388Met]	CD	M	Hidalgo	1529	cPKU	1.8
30	c. [1162G > A]; [1162G > A]	p. [Val388Met]; [Val388Met]	NBS	M	Queretaro	2265	mPKU	1.8
31	c. [1162G > A]; [1162G > A]	p. [Val388Met]; [Val388Met]	NBS	F	Queretaro	2258	mPKU	1.8
32	c. [194T > C]; [1241A > G]	p. [Ile65Thr]; [Tyr414Cys]	NBS	M	Zacatecas	790	mPKU	5.1
33	c. [194T > C]; [1241A > G]	p. [Ile65Thr]; [Tyr414Cys]	NBS	M	Queretaro	1070	mPKU	5.1
^♦^ 34	c. [441 + 5G > T]; [441 + 5G > T]	p. [?]; [?]	NBS	F	Tamaulipas	1640	mPKU	0
35	c. [441 + 5G > T]; [441 + 5G > T]	p. [?]; [?]	CD	M	Coahuila	1460	cPKU	0
^♦^ 36 ^Δ^	c. [969 + 6T > A]; [1162G > A]	p. [?]; [Val388Met]	NBS	F	CDMX	966	MHP	1.8
37 ^Δ^	c. [969 + 6T > A]; [1162G > A]	p. [?]; [Val388Met]	NBS	F	Jalisco	424	MHP	1.8
38	c. [1066-11G > A]; [1162G > A]	p. [Gln355_Tyr356insGlyLeuGln]; [Val388Met]	CD	F	CDMX	674	cPKU	1.8
39	c. [1066-11G > A]; [1162G > A]	p. [Gln355_Tyr356insGlyLeuGln]; [Val388Met]	NBS	F	Puebla	1304	mPKU	1.8
40	c. [1208C > T]; [1208C > T]	p. [Ala403Val]; [Ala403Val]	NBS	M	Queretaro	1748	MHP	9.7
41	c. [1208C > T]; [1208C > T]	p. [Ala403Val]; [Ala403Val]	NBS	F	CDMX	1477	MHP	9.7
^♦^ 42	c. [1A > T]; c. [1A > T]	p. [?]; [?]	CD	M	Veracruz	1431	cPKU	0
^♦^ 43 ^Δ^	c. [1A > T]; [722G > A]	p. [?]; [Arg241His]	NBS	M	Veracruz	1225	MHP	5.2
44	c. [1A > T]; [1042C > G]	p. [?]; [Leu348Val]	NBS	F	CDMX	1852	cPKU	1.5
45	c. [1A > T]; [1243G > A]	p. [?]; [Asp415Asn]	NBS	M	Veracruz	652	MHP	10
46	c. [**23del**]; [1162G > A]	p. [**Asn8Thrfs * 30**]; [Val388Met]	NBS	F	Chihuahua	1187	cPKU	Unknown
47	c. [117C > G]; [441 + 5G > T]	p. [Phe39Leu]; [?]	NBS	M	Chihuahua	1075	mPKU	0.2
48	c. [117C > G]; [1157A > G]	p. [Phe39Leu]; [Tyr386Cys]	NBS	F	Durango	1184	mPKU	0.2
49	c. [165del]; [194T > C]	p. [Phe55Leufs * 6]; [Ile65Thr]	NBS	M	Durango	1066	cPKU	1.1
50	c. [165T > G]; [208_210del]	p. [Phe55Leu]; [Ser70del]	NBS	M	Chihuahua	2825	MHP	8.2
51	c. [168 + 5G > C]; [1066-11G > A]	p. [?]; [Gln355_Tyr356insGlyLeuGln]	NBS	M	Chihuahua	1222	cPKU	0
52	c. [194T > C]; [208_210del]	p. [Ile65Thr]; [Ser70del]	NBS	M	Zacatecas	1791	cPKU	1
53 ^Δ^	c. [194T > C]; [533A > G]	p. [Ile65Thr]; [Glu178Gly]	NBS	F	Sonora	1527	MHP	7.6
54	c. [204A > T]; [527G > T]	p. [Arg68Ser]; [Arg176Leu]	NBS	M	CDMX	799	MHP	9.7
^♦^ 55	c. [204A > T]; [782G > A]	p. [Arg68Ser]; [Arg261Gln]	NBS	M	Queretaro	1187	mPKU	5.4
56	c. [204A > T]; [829T > G]	p. [Arg68Ser]; [Tyr277Asp]	NBS	M	Veracruz	1255	MHP	5.4
57	c. [208_210del]; [728G > A]	p. [Ser70del]; [Arg243Gln]	NBS	F	Nuevo León	1970	mPKU	0.7
^♦^ 58	c. [439C > T]; [1045T > C]	p. [Pro147Ser]; [Ser349Pro]	NBS	M	Jalisco	2098	cPKU	0
^♦^ 59	c. [439C > T]; [782G > A]	p. [Pro147Ser]; [Arg261Gln]	NBS	F	Michoacan	1923	mPKU	1.6
^♦^ 60	c. [441 + 5G > T]; [1045T > C]	p. [?]; [Ser349Pro]	NBS	F	Edo Mex	1452	cPKU	0
61	c. [441 + 5G > T]; [1055del]	p. [?]; [Gly352Valfs * 48]	NBS	M	CDMX	1756	cPKU	0
^♦^ 62	c. [441 + 5G > T]; [165del]	p. [?]; [Phe55Leufs * 6]	CD	F	Chihuahua	1634	cPKU	0
^♦^ 63	c. [441 + 5G > T]; [782G > A]	p. [?]; [Arg261Gln]	CD	M	CDMX	890	cPKU	1.6
64	c. [441 + 5G > T]; [791A > G]	p. [?]; [His264Arg]	NBS	F	Edo Mex	921	cPKU	0
65	c. [473G > A]; [1045T > C]	p. [Arg158Gln]; [Ser349Pro]	NBS	M	CDMX	1656	cPKU	0
66	c. [508C > G]; [1243G > A]	p. [His170Asp]; [Asp415Asn]	NBS	M	Jalisco	1098	MHP	10
^♦^ 67	c. [526C > T]; [1066-11G > A]	p. [Arg176 *]; [Gln355_Tyr356insGlyLeuGln]	CD	M	CDMX	1113	cPKU	0
68	c. [533A > G]; [1162G > A]	p. [Glu178Gly]; [Val388Met]	NBS	F	Hidalgo	502	MHP	7.6
69	c. [533A > G]; [1169A > G]	p. [Glu178Gly]; [Glu390Gly]	NBS	M	CDMX	179	MHP	7.6
70	c. [533A > G]; [533A > G]	p. [Glu178Gly]; [Glu178Gly]	CD	M	Queretaro	1277	MHP	7.6
71	c. [533A > G]; [809G > A]	p. [Glu178Gly]; [Arg270Lys]	NBS	F	Edo Mex	1154	MHP	7.6
72	c. [**625_626insC**]; [625_626insC]	p. [?]; [?]	NBS	M	Veracruz	1008	cPKU	Unknown
73	c. [673C > A]; [830A > G]	p. [Pro225Thr]; [Tyr277Cys]	NBS	M	Quintana Roo	1022	mPKU	0
74	c. [688G > A]; [754C > T]	p. [Val230Ile]; [Arg252Trp]	NBS	F	Veracruz	1608	MHP	10
75	c. [721C > T]; [1066-11G > A]	p. [Arg241Cys]; [Gln355_Tyr356insGlyLeuGln]	NBS	M	CDMX	1747	MHP	5.5
76	c. [722G > A]; [782G > A]	p. [Arg241His]; [Arg261Gln]	NBS	F	CDMX	2396	MHP	5.2
77 ^●^	c. [722G > A]; [842 + 1G > A]	p. [Arg241Cys]; [?]	NBS	M	Tabasco	2134	cPKU	5.2
78	c. [722G > A]; [722G > A]	p. [Arg241Cys]; [Arg241His]	NBS	M	Puebla	4627	MHP	5.2
79	c. [727C > T]; [1243G > A]	p. [Arg241Cys]; [Asp415Asn]	NBS	F	CDMX	948	MHP	10
80	c. [728G > A]; [1243G > A]	p. [Arg243Gln]; [Asp415Asn]	NBS	M	CDMX	712	MHP	10
^♦^ 81 ^Δ^	c. [728G > A]; [722G > A]	p. [Arg243Gln]; [Arg241His]	NBS	F	CDMX	506	MHP	5.2
82	c. [728G > A]; [809G > A]	p. [Arg243Gln]; [Arg270Lys]	NBS	M	Chihuahua	587	mPKU	0
^♦^ 83	c. [781C > T]; [1066-11G > A]	p. [Arg261 *]; [Gln355_Tyr356insGlyLeuGln]	CD	M	Jalisco	1623	cPKU	0
^♦^ 84	c. [781C > T]; [782G > A]	p. [Arg261 *]; [Arg261Gln]	NBS	F	CDMX	177	cPKU	1.6
85	c. [781C > T]; [1243G > A]	p. [Arg261 *]; [Asp415Asn]	NBS	M	CDMX	470	MHP	10
86	c. [782G > A]; [1045T > C]	p. [Arg261Gln]; [Ser349Pro]	NBS	M	Nuevo León	794	cPKU	1.6
87	c. [782G > A]; [1169A > G]	p. [Arg261Gln]; [Glu390Gly]	NBS	F	Guanajuato	215	MHP	7
88	c. [791A > G]; [**1315 + 5_1315 + 6insGTGTAACAG**]	p. [His264Arg]; [?]	NBS	M	Veracruz	401	cPKU	Unknown
89 ^Δ^	c. [809G > A]; [1162G > A]	p. [Arg270Lys]; [Val388Met]	NBS	M	San Luis Potosi	467	mPKU	1.8
90	c. [809G > A]; [1066-11G > A]	p. [Arg270Lys]; [Gln355_Tyr356insGlyLeuGln]	NBS	M	Puebla	282	cPKU	0
91	c. [814G > T]; [1282C > T]	p. [Gly272 *]; [Gln428 *]	NBS	F	Edo Mex	566	mPKU	0
92	c. [830A > G]; [830A > G]	p. [Tyr277Cys]; [Tyr277Cys]	NBS	M	Yucatan	454	MHP	0
^♦^ 93	c. [838G > A]; [838G > A]	p. [Glu280Lys]; [Glu280Lys]	NBS	F	CDMX	283	mPKU	0
94	c. [838G > A]; [1066-11G > A]	p. [Glu280Lys]; [Gln355_Tyr356insGlyLeuGln]	NBS	F	Colima	512	cPKU	0
^♦^ 95	c. [842 + 1G > A]; [441 + 5G > T]	p. [?]; [?]	NBS	M	Edo Mex	276	cPKU	0
^♦^ 96 ^●^	c. [842 + 1G > A]; [508C > G]	p. [?]; [His170Asp]	CD	M	Guanajuato	239	mPKU	10
97	c. [842 + 1G > A]; [932T > C]	p. [?]; [Leu311Pro]	CD	F	Sonora	239	cPKU	0
98	c. [898G > T]; [1162G > A]	p. [Ala300Ser]; [Val388Met]	NBS	F	Queretaro	289	MHP	9.7
^♦^ 99	c. [907T > G]; [673C > A]	p. [Ser303Ala]; [Pro225Thr]	NBS	F	CDMX	1351	MHP	9.3
^♦^ 100	c. [912G > A]; [912G > A]	p. [Gln304=]; [Gln304=]	CD	F	Sinaloa	218	cPKU	3.3
^♦^ 101	c. [941C > A]; [809G > A]	p. [Pro314His]; [Arg270Lys]	NBS	M	Aguascalientes	245	MHP	10
102	c. [969 + 6T > A]; [1198A > C]	p. [?]; [Arg400=]	NBS	M	Michoacan	278	MHP	Undetermined
^♦^ 103	c. [1042C > G]; [194T > C]	p. [Leu348Val]; [Ile65Thr]	CD	M	Morelos	180	mPKU	1.5
104	c. [1042C > G]; [1238G > A]	p. [Leu348Val]; [Arg413His]	NBS	F	Tabasco	248	cPKU	9
^♦^ 105	c. [1045T > C]; [1066-11G > A]	p. [Ser349Pro]; [Gln355_Tyr356insGlyLeuGln]	NBS	M	Chihuahua	242	cPKU	0
^♦^ 106	c. [1045T > C]; [1169A > G]	p. [Ser349Pro]; [Glu390Gly]	NBS	F	CDMX	600	MHP	7
107	c. [1045T > C]; [1208C > T]	p. [Ser349Pro]; [Ala403Val]	NBS	F	Guerrero	455	MHP	9.7
^♦^ 108	c. [1045T > C]; [472C > T]	p. [Ser349Pro]; [Arg158Trp]	CD	M	CDMX	515	cPKU	0
109	c. [1045T > C]; [1162G > A]	p. [Ser349Pro]; [Val388Met]	NBS	F	Chihuahua	366	mPKU	1.8
110	c. [1045T > C]; [1243G > A]	p. [Ser349Pro]; [Asp415Asn]	NBS	M	Veracruz	425	MHP	10
111	c. [1055del]; [1055del]	p. [Gly352Valfs * 48]; [Gly352ValfsTer48]	NBS	F	Hidalgo	537	cPKU	0
112	c. [1055del]; [1162G > A]	p. [Gly352Valfs * 48]; [Val388Met]	NBS	M	Hidalgo	498	cPKU	1.8
^♦^ 113	c. [1066-11G > A]; [.809G > A]	p. [Gln355_Tyr356insGlyLeuGln]; [Arg270Lys]	NBS	M	Chihuahua	654	mPKU	0
^♦^ 114	c. [1066-11G > A]; [969 + 6T > C]	p. [Gln355_Tyr356insGlyLeuGln]; [?]	NBS	M	Chiapas	453	cPKU	Undetermined
^♦^ 115 ^Δ^	c. [1127delA]; [1066-11G > A]	p. [Asn376Ilefs * 24]; [Gln355_Tyr356insGlyLeuGln]	CD	M	Veracruz	244	mPKU	Undetermined
^♦^ 116 ^Δ^	c. [1157A > G]; [754C > T]	p. [Tyr386Cys]; [Arg252Trp]	CD	F	Oaxaca	365	mPKU	0
117	c. [116_118del]; [1045T > C]	p. [Phe39del]; [Ser349Pro]	NBS	F	CDMX	377	cPKU	0.9
^♦^ 118	c. [1162G > A]; [1066-11G > A]	p. [Val388Met]; [Gln355_Tyr356insGlyLeuGln]	NBS	M	Guerrero	628	cPKU	1.9
119	c. [1162G > A]; c. [1243G > A]	p. [Val388Met]; [Asp415Asn]	NBS	F	Michoacan	2798	MHP	10
^♦^ 120	c. [1162G > A]; [1315 + 1G > A]	p. [Val388Met]; [?]	CD	F	Edo Mex	1217	cPKU	1.9
^♦^ 121	c. [1169A > G]; [1169A > G]	p. [Glu390Gly]; [Glu390Gly]	NBS	F	Michoacan	1753	MHP	7
122	c. [1243G > A]; [1315 + 1G > A]	p. [Asp415Asn]; [?]	NBS	M	Tabasco	192	MHP	10
^♦^ 123 ^●^	c. [1315 + 1G > A]; [508C > G]	p. [?]; [His170Asp]	NBS	M	Guanajuato	1779	mPKU	10
^♦^ 124 ^●^	c. [1315 + 2T > C]; [1162G > A]	p. [?]; [Val388Met]	CD	F	Queretaro	1060	cPKU	1.8

Novel pathogenic *PAH* variants are highlighted in red bold type. GPV is equal to the higher APV of the two alleles (APV max). The GPV cut-off values are defined as cPKU (0–2.7), mPKU (2.8–6.6) and MHP (6.7–10.0). Abbreviations: CD, clinical diagnosis; cPKU, classic phenylketonuria; F, female; M, male; MHP, mild hyperphenylalaninemia; mPKU, mild phenylketonuria; NBS, newborn screening; NR, not reported or still undetermined. Symbols represent genotype/phenotype discordance, **^Δ^** underestimation of disease severity, and ^●^ overestimation of disease severity compared with BIOPKUdb. ^♦^ These patients were described in our previous report [6].

**Table 3 genes-12-01676-t003:** Main biochemical and clinical characteristics of the patients harboring the c. 60 + 5G < T variant.

Patient ID	*PAH* Genotype	Observed Biochemical Phenotype	GPV	Diagnosis	Max Historical Phe (µmol/L)	Tyr (µmol/L)	Phe/Tyr Ratio
1	c. [60 + 5G > T]; [60 + 5G > T]	cPKU	0.0	NBS	3591	70	41.26
2	c. [60 + 5G > T]; [60 + 5G > T]	cPKU	0.0	CD	2421	64	37.83
3	c. [60 + 5G > T]; [60 + 5G > T]	cPKU	0.0	CD	1896	38	49.89
4	c. [60 + 5G > T]; [60 + 5G > T]	cPKU	0.0	CD	1654	38	43.53
5	c. [60 + 5G > T]; [60 + 5G > T]	cPKU	0.0	CD	1876	42	44.67
6	c. [60 + 5G > T]; [60 + 5G > T]	cPKU	0.0	NBS	1573	61	25.79
7	c. [60 + 5G > T]; [60 + 5G > T]	cPKU	0.0	NBS	1311	33	39.73
8	c. [60 + 5G > T]; [60 + 5G > T]	cPKU	0.0	NBS	1712	39	43.90
9	c. [60 + 5G > T]; [60 + 5G > T]	cPKU	0.0	NBS	1581	35	45.17
10	c. [60 + 5G > T]; [60 + 5G > T]	cPKU	0.0	NBS	1453	109	13.33
11	c. [60 + 5G > T]; [1162G > A]	cPKU	1.8	CD	1923	124	15.51
12	c. [60 + 5G > T]; [1162G > A]	cPKU	1.8	NBS	1452	26	55.85
13	c. [60 + 5G > T]; c. [1162G > A]	cPKU	1.8	NBS	1756	38	46.21
14	c. [60 + 5G > T]; [1162G > A]	cPKU	1.8	NBS	1634	106	15.42
15	c. [60 + 5G > T]; [441 + 5G > T]	cPKU	0.0	NBS	1900	36	52.78
16	c. [60 + 5G > T]; [441 + 5G > T]	cPKU	0.0	NBS	3832	78	49.13
17	c. [60 + 5G > T]; [441 + 5G > T]	cPKU	0.0	CD	1876	58	32.34
18	c. [60 + 5G > T]; [1315 + 1G > A]	cPKU	0.0	NBS	1715	46	37.28
19	c. [60 + 5G > T]; [1315 + 1G > A]	cPKU	0.0	NBS	1682	70	24.03
20	c. [60 + 5G > T]; [673C > A]	cPKU	0.0	NBS	1815	69	26.30
21	c. [60 + 5G > T]; [727C > T]	cPKU	0.0	CD	1346	108	12.46
22	c. [60 + 5G > T]; [1066-11G > A]	cPKU	0.0	NBS	1385	113	12.26
23	c. [60 + 5G > T]; [1169A > G]	MHP	6.9	NBS	467	103	4.53
24	c. [ 60 + 5G > T]; [ 508C > G]	MHP	10.0	NBS	600	77	7.79
25	c. [ 60 + 5G > T]; [ 754C > T]	cPKU	0.0	CD	1206	33	36.55
26	c. [ 60 + 5G > T]; [ 838G > A]	cPKU	0.0	CD	2326	41	56.73

Abbreviations: cPKU, classic phenylketonuria; NBS, newborn screening; CD, clinical diagnosis; GVP, genotypic phenotype value; MHP, mild hyperphenylalaninemia; mPKU, mild phenylketonuria; Phe, phenylalanine; Tyr, tyrosine.

**Table 4 genes-12-01676-t004:** Potential BH_4_-responder patients (65/124, 52%) according to their genotype, and those who were recommended for the BH_4_-responsiveness test.

Genotype		% BH_4_ Response Reported in BIOPKU		
	Observed Biochemical Phenotype	Allele 1 in Homozygous State Yes/Slow	Allele 2 in Homozygous State Yes/Slow	BH_4_ Responsiveness
c. [722G > A]; [722G > A]	MHP	100/0	100/0	Identical genotype reported in BIOPKUdb as responder
c. [1169A > G]; [1169A > G]	MHP	100/0	100/0	
c. [1208C > T]; [1208C > T]	MHP	100/0	100/0	
c. [722G > A]; [782G > A]	MHP	100/0	74/4	
c. [722G > A]; [842 + 1G > A]	cPKU	100/0	0	
c. [782G > A]; [1169A > G]	MHP	74/4	100/0	
c. [728G > A]; [1243G > A]	MHP	0/6.67	NR	
c. [60 + 5G > T]; [1169A > G]	MHP	NR	100/0	
c. [1045T > C]; [1169A > G]	MHP	0	100/0	
c. [1045T > C]; [1208C > T]	MHP	0	100/0	
c. [1045T > C]; [1243G > A]	MHP	0	NR	
c. [60 + 5G > T]; [508C > G]	MHP	NR	NR	
c. [204A > T]; [527G > T]	MHP	100/0	NR	Potential responder
c. [204A > T]; [782G > A]	mPKU	100/0	74/4	
c. [204A > T]; [829T > G]	MHP	100/0	0	
c. [721C > T]; [1066-11G > A]	MHP	100/0	0/6.09	
c. [722G > A]; [728G > A]	MHP	100/0	0/6.67	
c. [1042C > G]; [194T > C]	mPKU	100/0	84/0	
c. [1042C > G]; [1238G > A]	cPKU	100/0	NR	
c. [194T > C]; [208_210del]	cPKU	84/0	NR	
c. [194T > C]; [533A > G]	MHP	84/0	NR	
c. [194T > C]; [1241A > G]	mPKU	84/0	100/0	
c. [782G > A]; [1045T > C]	cPKU	74/4	0	
c. [1162G > A]; [1162G > A]	mPKU	62.5/12.5	62.5/12.5	
c. [1162G > A]; c. [1243G > A]	MHP	62.5/12.5	NR	
c. [1162G > A]; [1315 + 1G > A]	cPKU	62.5/12.5	0/13	
c. [1162G > A]; [1315 + 2T > C]	cPKU	62.5/12.5	NR	
c. [898G > T]; [1162G > A]	MHP	60/40	62.5/12.5	
c. [1055del]; [1162G > A]	cPKU	5/5	62.5/12.5	
c. [116_118del]; [1045T > C]	cPKU	0/80	0	
c. [117C > G]; [1157A > G]	mPKU	0/50	NR	
c. [117C > G]; [441 + 5G > T]	mPKU	0/50	NR	
c. [781C > T]; [782G > A]	cPKU	0/20	74/4	
c. [781C > T]; [1243G > A]	MHP	0/20	NR	
c. [1066-11G > A]; [1127delA]	mPKU	0/6.09	NR	
c. [1066-11G > A]; [1162G > A]	cPKU	0/6.09	62.5/12.5	
c. [673C > A]; [907T > G]	MHP	0	NR	
c. [1045T > C]; [1162G > A]	mPKU	0	62.5/12.5	
c. [1A > T]; [722G > A]	MHP	NR	100/0	
c. [1A > T]; [1042C > G]	cPKU	NR	100/0	
c. [**23del**]; [1162G > A]	cPKU	NR	62.5/12.5	
c. [60 + 5G > T]; [1162G > A]	cPKU	NR	62.5/12.5	
c. [165del]; [194T > C]	cPKU	NR	84/0	
c. [165T > G]; [208_210del]	MHP	NR	NR	
c. [208_210del]; [728G > A]	mPKU	NR	0/6.67	
c. [439C > T]; [782G > A]	mPKU	NR	74/4	
c. [441 + 5G > T]; [782G > A]	cPKU	NR	74/4	
c. [508C > G]; [842 + 1G > A]	mPKU	NR	0	
c. [508C > G]; [1243G > A]	MHP	NR	NR	
c. [508C > G]; [1315 + 1G > A]	mPKU	NR	0/13	
c. [533A > G]; [1162G > A]	MHP	NR	62.5/12.5	
c. [533A > G]; [1169A > G]	MHP	NR	100/0	
c. [791A > G]; [**1315 + 5_1315 + 6insGTGTAACAG**]	cPKU	NR	NR	
c. [809G > A]; [941C > A]	MHP	NR	NR	
c. [809G > A]; [1162G > A]	mPKU	NR	62.5/12.5	
c. [969 + 6T > C]; [1066-11G > A]	cPKU	NR	0/6.09	
c. [969 + 6T > A]; [1162G > A]	MHP	NR	62.5/12.5	
c. [1243G > A]; [1315 + 1G > A]	MHP	NR	0/13	
c. [1A > T]; [1243G > A]	MHP	NR	NR	Undetermined response
c. [533A > G]; [533A > G]	MHP	NR	NR	
c. [533A > G]; [809G > A]	MHP	NR	NR	
c. [**625_626insC**]; [**625_626insC**]	cPKU	NR	NR	
c. [688G > A]; [754C > T]	MHP	NR	0	
c. [727C > T]; [1243G > A]	MHP	No	NR	
c. [969 + 6T > A]; [1198A > C]	MHP	NR	NR	

Novel pathogenic *PAH* variants are highlighted in red bold type. Abbreviations: cPKU, classic phenylketonuria; MHP, mild hyperphenylalaninemia; mPKU, mild phenylketonuria; NR, not reported or still undetermined.

## Data Availability

The datasets analyzed during the present study are available from the corresponding author on reasonable request.

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
