# Peer review of "An Updated PAH Mutational Spectrum of Phenylketonuria in Mexican Patients Attending a Single Center: Biochemical, Clinical-Genotyping Correlations"

_genes, 2021, doi:10.3390/genes12111676_

Round 1
Reviewer 1 Report
The manuscript deals with the mutation spectrum of a well-described disease, where it is difficult to bring any new knowledge, but as the authors state, it is an update of the PAH mutation spectrum published in the past by the authors. From this point of view, this information should be rather presented as entries in the databases. The authors identified together only 3 new mutations, which bring very little new knowledge. On the other hand, the paper brings comprehensive information (partly already published) about PAH mutations in the Mexican population.
Major revisions
The manuscript needs to be revised to remove overrepresented data. Many of the presented results are only taken from the BioPKU database. Although it provides a comprehensive view of certain clinical indicators, it is not necessary to state and repeat it in up to 6 Tables. Tables 2 and 3 can be joined together with the indication of the relevant information from Table 3. In Table 6, there is nothing new, information is already in Table 1 and deducible from Table 5.
I also don't know if the comparison of mutation frequencies with the bioPKU database is meaningful and authors make inappropriate conclusions. Not all patients from different populations are listed in database, often one-two is reported for the first time with a given mutation, but not every PAH patient. Correctly, data should be compared with other population-oriented mutation spectrum studies. In addition, those frequencies you compare are a mix of different populations' frequencies, which is also skewed depending on how many records are from each country.
You mentioned: “novel unclassified, novel and missense changes were tested in silico using Polyphen, …..”. Although the authors report in silico predictive testing, it is not stated which novel, unclassified and missense variants were tested and no results from these tests are presented. The new variants are only insertion/deletion variants, but the result of variants testing is also not presented.
The discussion is long and often repeats the results, it could be shorter
Minor revisions Definition in Material and methods for GPV, APV, enzymatic activity is not necessary it is not methods which authors developed or used. Not precisely defined methods – they use Purgene kit- it is a group of kits. You used gentra puregene kit, blood or tissue puregene kit?? Should be stated
Check grammar and prepositions
MLPA instead of MPLA
Author Response
Manuscript title: “An updated PAH mutational spectrum of phenylketonuria in Mexican patients attending a single center: biochemical, clinical-genotyping correlations”. Manuscript ID: genes-1418497.
Please find below the response to each of your comments. Changes made to the manuscript were marked using the “Track Changes” function. We thank the reviewer for his/her comment which improved our manuscript.
Reviewer #1:
The manuscript deals with the mutation spectrum of a well-described disease, where it is difficult to bring any new knowledge, but as the authors state, it is an update of the PAH mutation spectrum published in the past by the authors. From this point of view, this information should be rather presented as entries in the databases. The authors identified together only 3 new mutations, which bring very little new knowledge. On the other hand, the paper brings comprehensive information (partly already published) about PAH mutations in the Mexican population.
Major revisions
- The manuscript needs to be revised to remove overrepresented data. Many of the presented results are only taken from the BioPKU database. Although it provides a comprehensive view of certain clinical indicators, it is not necessary to state and repeat it in up to 6 Tables. Tables 2 and 3 can be joined together with the indication of the relevant information from Table 3. In Table 6, there is nothing new, information is already in Table 1 and deducible from Table 5.
RESPONSE: Thank you for your valuable observations. In the new version of the manuscript, we joined Tables 2 and 3. As suggested, Table 6 was deleted.
- I also don't know if the comparison of mutation frequencies with the bioPKU database is meaningful and authors make inappropriate conclusions. Not all patients from different populations are listed in database, often one-two is reported for the first time with a given mutation, but not every PAH patient. Correctly, data should be compared with other population-oriented mutation spectrum studies. In addition, those frequencies you compare are a mix of different populations' frequencies, which is also skewed depending on how many records are from each country.
RESPONSE: Thank you for this wise comment. We agree with the reviewer’s opinion. However, BIOPKUdb contains representative data of many PKU patients worldwide, therefore we used it as a reference. Certainly, the Latin American population is underrepresented, so we also analyzed the published reports of Latin American population in order to compare our findings.
In order to clarify, we added the following sentences as a limitation of our study:
Page 34: “One limitation for the frequency of variants and concordance comparison is that not all patients from different populations are listed in BIOPKUdb.”
- You mentioned: “novel unclassified, novel and missense changes were tested in silico using Polyphen, …..”. Although the authors report in silico predictive testing, it is not stated which novel, unclassified and missense variants were tested and no results from these tests are presented. The new variants are only insertion/deletion variants, but the result of variants testing is also not presented.
RESPONSE: We appreciate this observation. In fact, this in silico evaluation was proposed in case of identify any unreported missense change, a situation that we did not identify in our study. However, due to the limited information available to date for the p.(His264Arg) variant, this in silico evaluation was carried-out before to perform the protein modeling. Then, to address this request, we modified the sentence allocated in Material and Methods (section 2.7, page 4) as follows:
“Due to limited available information regarding their functional or clinical relevance, these unclassified novel and missense changes were subjected to in silico analysis using the PolyPhen (http://genetics.bwh.harvard.edu/pph2, accessed on 20 December 2020), PROVEAN (http://provean.jcvi.org/index.php, accessed on 20 December 2020), and Pmut (http://mmb.irbbarcelona.org/PMut, accessed on 20 December 2020) programs.”
Also, we include the results evaluations of PolyPhen, PROVEAN, and Pmut programs for p.(Arg264His) variant in section 3.7 (RESULTS):
3.7. Protein in silico modeling of the p.(His264Arg) variant
The c.791A>G or p.(His264Arg) has been recently reported in the BIOPKUdb, although its allele frequency is still not known, results of functional assays are not available, and its resulting biochemical phenotype is still unknown. In fact, in the ClinVar database, it is considered as a variant of unknown significance (RCV001224584.2). Herein, it was identified in two unrelated patients (ID 64 and 88, Table 2). The evaluation of by the PolyPhen, PROVEAN, and Pmut programs unanimously predicted the p.(His264Arg) variant as deleterious with high confidence scores. Due to the rarity of this variant, we evaluated in silico the potential disturbance at the tridimensional structural arrangement of the resulting protein. We found that the substitution of histidine by the positively charged amino acid arginine predicts a serious disturbing effect within the vicinity of the catalytic site in all the possible rotamers of arginine (Figure 5).
- The discussion is long and often repeats the results, it could be shorter
RESPONSE: We have shortened the discussion, omitting repetitive information.
- Minor revisions.
Definition in Material and methods for GPV, APV, enzymatic activity is not necessary it is not methods which authors developed or used.
RESPONSE: We apologize for this mistake. We deleted these sections from Material and methods. Instead GPV and APV abbreviations were stated at the table legends of the corresponding tables.
6a. Not precisely defined methods – they use Purgene kit- it is a group of kits. You used gentra puregene kit, blood or tissue puregene kit?? Should be stated
RESPONSE: We appreciate this observation. Then, the sentence of Material and Methods section 2.7 was modified as follows:
“Genomic DNA samples were obtained from dried blood spots by the saline precipitation method (Gentra Puregene Blood Kit; Gentra Systems, Minneapolis, MN, USA).”
6b. Check grammar and prepositions
RESPONSE: Thank you for the observation. The manuscript was sent to the Language Editing Services from MDPI.
- MLPA instead of MPLA.
RESPONSE: The mistake was corrected on page 17.

Reviewer 2 Report
This is a well-done manuscript that brings new information regarding a PAH Mutational Spectrum of Phenylketonuria in Mexican Patients Attending a Single Center: Biochemical, Clinical-Genotyping Correlation
Abstract is presented in a clear, succinct and direct way. Introduction is clear, containing all the necessary information to understand the objectives of this study. Methodology is clear and straightforward. Results and discussion are clear and in accordance with the objective. However, some aspects need to be improved in the manuscript.
The title in Table 2 is somewhat mistaken, as there is more information than simply the genotype of the Mexican HPA/PKU patients studied. In this table I checked information on diagnosis, sex, geographic origin and biochemical phenotype observed. This table is also very extensive. I suggest that table 2 be replaced by a text informing these variables, since mutations are somewhat mentioned in other tables of the article.
Brazil is a country divided into five geographical regions: north, northeast, midwest, southeast and south. Only indigenous peoples inhabited the brazilian territory before the 16th century. From the 16th century, Europeans and black Africans (enslaved) arrived in Brazil. Thus, the genetic background of the Brazilian people was constituted. There are several studies on the molecular biology of PKU in Brazil.
References 20 and 25 used in the article in question do not fully represent the studies that were done in Brazil. In addition, reference 25 was poorly applied (line 07 on page 18), since the study done in south Brazil was: Molecular characterization of phenylketonuria in South Brazil. Mol Genet Metab. 2003 May;79(1):17-24. doi: 10.1016/s1096-7192(03)00032-5.
To support the evidence of this article, I also suggest reading these other Brazilian articles:
Phenylketonuria Diagnosis by Massive Parallel Sequencing and Genotype-Phenotype Association in Brazilian Patients. Tresbach et al. Genes (Basel). 2020 Dec 25;12(1):20. doi: 10.3390/genes12010020.
PKU in Minas Gerais State, Brazil: mutation analysis. L.L. Santos et. al. Ann Hum Genet. 2008 Nov;72(Pt 6):774-9. doi: 10.1111/j.1469-1809.2008.00476.x. Epub 2008 Sep 16.
Author Response
Manuscript title: “An updated PAH mutational spectrum of phenylketonuria in Mexican patients attending a single center: biochemical, clinical-genotyping correlations”. Manuscript ID: genes-1418497.
Please find below the response to each of your comments. Changes made to the manuscript were marked using the “Track Changes” function. We thank the reviewer for his/her comment which improved our manuscript.
Reviewer #2:
This is a well-done manuscript that brings new information regarding a PAH Mutational Spectrum of Phenylketonuria in Mexican Patients Attending a Single Center: Biochemical, Clinical-Genotyping Correlation.
Abstract is presented in a clear, succinct, and direct way. Introduction is clear, containing all the necessary information to understand the objectives of this study. Methodology is clear and straightforward. Results and discussion are clear and in accordance with the objective. However, some aspects need to be improved in the manuscript.
- The title in Table 2 is somewhat mistaken, as there is more information than simply the genotype of the Mexican HPA/PKU patients studied. In this table I checked information on diagnosis, sex, geographic origin and biochemical phenotype observed. This table is also very extensive. I suggest that table 2 be replaced by a text informing these variables, since mutations are somewhat mentioned in other tables of the article.
RESPONSE: We agree. In the new version of the manuscript, to describe accurately the content of table 2, the title was modified as follows:
“Table 2. Demographic, clinical, and biochemical characteristics of the studied Mexican HPA/PKU patients (N=124).”
Also, table 2 was reconstructed as suggested by both reviewers. Tables 2 and 3 were joined in order to simplify the information.
- Brazil is a country divided into five geographical regions: north, northeast, midwest, southeast and south. Only indigenous peoples inhabited the brazilian territory before the 16th century. From the 16th century, Europeans and black Africans (enslaved) arrived in Brazil. Thus, the genetic background of the Brazilian people was constituted. There are several studies on the molecular biology of PKU in Brazil.
RESPONSE: Thank you for this pertinent observation. We recognize the complexity of Brazilian trihybrid genetic background. We attenuated generalizations to Brazilian population.
- References 20 and 25 used in the article in question do not fully represent the studies that were done in Brazil. In addition, reference 25 was poorly applied (line 07 on page 18), since the study done in south Brazil was: Molecular characterization of phenylketonuria in South Brazil. Mol Genet Metab. 2003 May;79(1):17-24. doi: 10.1016/s1096-7192(03)00032-5.
RESPONSE: Thank you for this observation. In the new version of the manuscript the misleading sentence where reference 25 was poorly applied, was eliminated.
- To support the evidence of this article, I also suggest reading these other Brazilian articles:
Phenylketonuria Diagnosis by Massive Parallel Sequencing and Genotype-Phenotype Association in Brazilian Patients. Tresbach et al. Genes (Basel). 2020 Dec 25;12(1):20. doi: 10.3390/genes12010020.
PKU in Minas Gerais State, Brazil: mutation analysis. L.L. Santos et. al. Ann Hum Genet. 2008 Nov;72(Pt 6):774-9. doi: 10.1111/j.1469-1809.2008.00476.x. Epub 2008 Sep 16.
RESPONSE: Thank you for this observation. We would like to note that the reference “Phenylketonuria Diagnosis by Massive Parallel Sequencing and Genotype-Phenotype Association in Brazilian Patients. Tresbach et al. Genes (Basel). 2020 Dec 25;12(1):20. doi: 10.3390/genes12010020” was already included in the original version of the manuscript (page 25, reference number 34).
As suggested, we revised and integrated the reference “PKU in Minas Gerais State, Brazil: mutation analysis. L.L. Santos et. al. Ann Hum Genet. 2008 Nov;72(Pt 6):774-9. doi: 10.1111/j.1469-1809.2008.00476.x. Epub 2008 Sep 16”.
